# Examining causal pathways between family adversity and soiling: a prospective cohort study

Kimberley Burrows ![ORCID],[1] Jon Heron ![ORCID],[2,3] Davina Richardson,[4] Juliette Rayner,[5] Christine Spray ![ORCID],[6] Carol Joinson[1]

For numbered affiliations see end of article.

**Correspondence to**
Dr Kimberley Burrows; kimberley.burrows@bristol.ac.uk

## ABSTRACT

**Objective** We aimed to examine the associations between family adversity in infancy and subsequent daytime soiling at primary school age. We also examined factors on the causal pathway between family adversity and soiling, specifically the child's emotional/behaviour problems and constipation.

**Design** This prospective cohort study included children from the Avon Longitudinal Study of Parents and Children (N=10 033) with data on the Family Adversity Index (FAI; birth to age 2), daytime soiling at age 7.5, emotional/behaviour problems at age 4, and constipation at age 6. We examined the relationships between the FAI, soiling, emotional/behaviour problems and constipation. We then evaluated the mediating effects of emotional/behaviour problems and constipation. All analyses were adjusted for child and family-related confounders.

**Results** Daytime soiling occurred in 7% of children. A one-unit increase in infancy FAI was associated with increased odds of 12% for daytime soiling at age 7.5 (OR 1.12, 95% CI 1.07 to 1.17), emotional/behaviour problems score at age 4 (beta 0.52, 95% CI 0.47 to 0.58) and increased odds of constipation at age 6 (OR 1.07, 95% CI 1.03 to 1.11). The presence of constipation at age 6 was strongly associated with increased odds of soiling at age 7.5 (OR 3.34, 95% CI 2.68 to 4.16). There was weak evidence that the associations between FAI and daytime soiling were mediated by emotional/behaviour problems and constipation ($OR_{natural\_indirect\_effect}$ 1.03, 95% CI 0.99 to 1.07).

**Conclusions** Clinicians should be aware that exposure to family adversity in the early years places children at greater risk of subsequent constipation and soiling.

## INTRODUCTION

Soiling (faecal incontinence) is a common paediatric health problem (estimated prevalence of 4% in primary school-age children),[1] which causes significant distress to children and their parents[2] as well as presenting challenges within educational settings. Most cases of childhood soiling are functional and are a consequence of overflow due to chronic constipation.[3] Stool withholding and toileting refusal, due to early painful experiences of passing hard stools, are primary causes of functional constipation in children.[3]

## WHAT IS ALREADY KNOWN ON THIS TOPIC

⇒ There is growing evidence that adversities including exposure to stressful life events, maternal psychopathology and family discord may play a role in determining a child's later risk of soiling.

## WHAT THIS STUDY ADDS

⇒ This is the first study to examine mechanisms explaining the relationship between adversity and soiling using causal mediation methods in a large community-based birth cohort study.
⇒ The total burden of family adversity (encompassing facets of psychopathology, discord and events) during infancy increases the odds of soiling at primary school age.
⇒ There is weak evidence that emotional/behaviour problems and constipation mediate the relationship between family adversity and soiling.

## HOW THIS STUDY MIGHT AFFECT RESEARCH, PRACTICE OR POLICY

⇒ Clinicians should be aware that exposure to family adversity in the early years places children at greater risk of subsequent soiling.
⇒ Increased resources need to be directed at providing support and interventions for families who are experiencing adversities to reduce the impact on children's health and development.

There is growing evidence that adversity in the child's early family environment plays a role in determining a child's later risk of soiling. Previous studies have focused on specific types of adversities including exposure to stressful life events, maternal psychopathology and family discord,[4–8] but none have examined the total burden of exposure to a range of adversities in the early years, and many were cross-sectional or used a retrospective design (reviewed in refs. 4 8). There is also a lack of research examining mechanisms that could explain how early adverse exposures could increase a child's subsequent risk of soiling.

The current paper, based on data from a large UK birth cohort, examines the

prospective relationship between exposure to family adversity in infancy (0–2 years) and subsequent soiling at primary school age (7.5 years). We use a measure of family adversity which comprises multiple domains including deprivation, low parental education, parental discord, parental crime, parental psychopathology and parents' lack of social/practical/emotional support. We also examine factors on the causal pathway between family adversity and soiling, specifically the child's emotional/behaviour problems and constipation. We chose to focus on these factors because these are potentially modifiable and could be targeted in interventions aimed at reducing the risk of soiling.[6 9–13]

## METHODS

We used the Strengthening the Reporting of Observational Studies in Epidemiology (STROBE) cohort reporting guidelines (see STROBE checklist in the online supplemental text).

### Sample

The Avon Longitudinal Study of Parents and Children (ALSPAC) is a prospective, population-based birth cohort study that recruited pregnant women resident in Avon, UK with expected dates of delivery between 1 April 1991 and 31 December 1992.[14 15] The initial number of pregnancies enrolled was 14 541 resulting in 14 062 live

births and 13 988 children who were alive at 1 year of age. Further details are given in the online supplemental text.

The eligible sample for this study included children who had available data on the family adversity index score (FAI; see below) derived for the period between birth and 2 years of age (N=10 033). Complete data for the FAI, mediators, daytime soiling and confounders were available for 3366 children. The participant flow chart is presented in figure 1.

### Patient and public involvement

Patients and/or the public were not involved in the design, conduct, reporting or dissemination of this research.

### Exposure: FAI

Multiple family, parental, sociodemographic and social support indicators were assessed via questionnaires completed during pregnancy and when the child was 0–2 years old. The FAI consisted of 18 items about the age of the mother at pregnancy, housing, parents' education, financial difficulties, relationship status and support, family size and care, social network, substance abuse, crime and psychopathology of parents. Total FAI score was derived by summing all 18 items, with higher scores indicating greater family adversity. See online supplemental text and online supplemental table 1 for further information on the FAI.

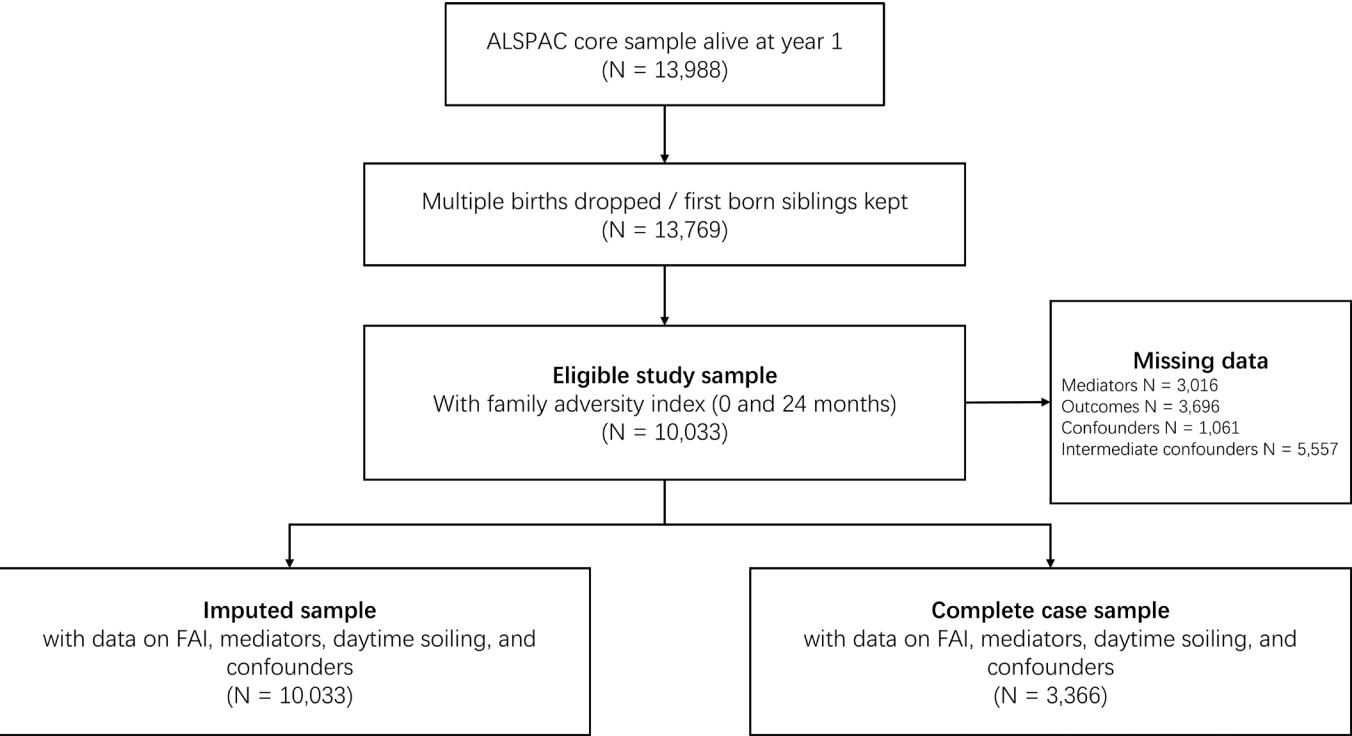

**Figure 1** Flow chart of inclusion and exclusion of ALSPAC participants into the study sample. Mediators include emotional/behaviour problems and constipation. Confounders include sex, ethnicity, preterm birth, household social class, marital status and smoking during pregnancy. Intermediate confounders include child developmental delay, maternal bond score, parental abuse, stool frequency, hard stools, difficult temperament, picky eating, healthy eating, temper tantrums and BMI. ALSPAC, Avon Longitudinal Study of Parents and Children; BMI, body mass index; FAI, Family Adversity Index.

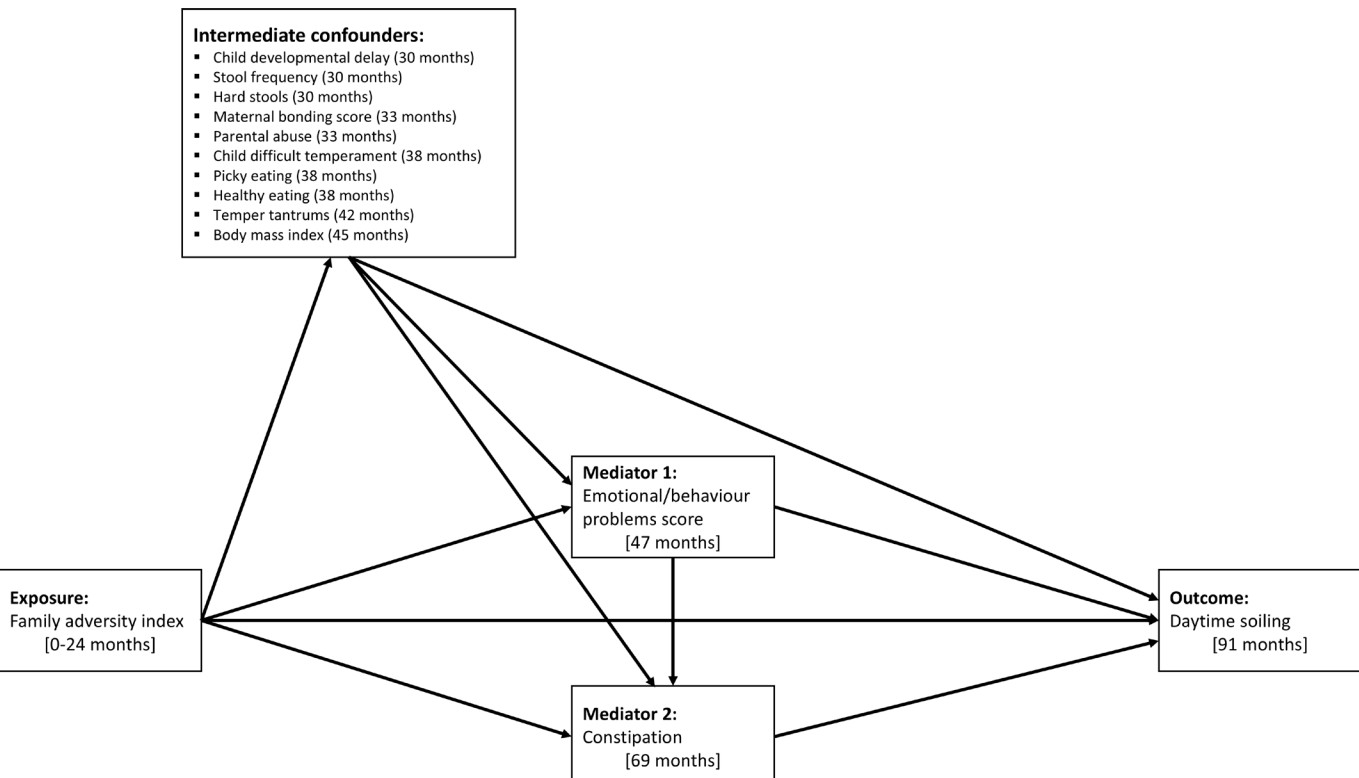

**Figure 2** Directed Acyclic Graph (DAG) illustrating hypothesised causal relationships between the Family Adversity Index, mediators and daytime soiling. Each intermediate confounding path was specified separately but is grouped together for clarity. For simplicity, baseline confounders are omitted from the DAG as all confounders are proposed to cause all other model variables.

## Outcomes: daytime soiling

When the study children were 7 years 8 months (SD=0.14) (hereafter referred to as 7.5 years), mothers were asked 'how often does your child dirty their pants during the day'. Options included 'never'; 'occasional accidents but less than once a week'; 'about once a week'; '2–5 times a week'; 'nearly every day' and 'more than once a day'. After exploring the proportions of respondents for the more frequent soiling categories (only 0.83% responded with 2–5 times a week or more), we chose to derive a binary variable to indicate any daytime soiling by assigning a value of 0 to 'never' and 1 to all other categories. Further information for coding daytime soiling is given in online supplemental table 1.

## Mediators: emotional/behavioural problems and constipation

We chose emotional/behavioural problems and constipation as mediators as these are potentially modifiable and could be targeted in interventions aimed at reducing the risk of soiling.[6 9–13] Furthermore, there is evidence that emotional/behavioural problems are associated with constipation and soiling,[6 16] and so we considered the two mediators jointly within our causal mediation models (see figure 2).

When the study child was 4 years (SD=0.12), mothers completed the Strengths and Difficulties Questionnaire (SDQ),[17] which comprises five subscales (each with five items: hyperactivity, conduct problems, emotional symptoms, peer problems and prosocial behaviour). An emotional/behaviour problems score was calculated from the SDQ using the first four subscales (range: 0–40). A high score indicates more emotional/behaviour problems.

When the study child was 5 years 10 months (SD=0.12) (hereafter referred to as 6 years), mothers were asked, 'Has your child had any constipation in the past 12 months'. Options included 'yes and saw a doctor'; 'yes but did not see a doctor'; 'no, did not have'. We derived a binary variable to indicate constipation by assigning a value of 0 to 'no, did not have' and 1 to the other two categories. Further information for coding the mediators is in online supplemental table 1.

## Baseline and intermediate confounders

We adjusted for baseline confounders (measured during the antenatal period or at birth) of each of the mediating pathways (exposure-outcome, exposure-mediator, and mediator-outcome). Confounders measured during the antenatal period included ethnicity, household social class, marital status and smoking during pregnancy. The confounders sex assigned at birth and preterm birth were measured during the postnatal period. Other indicators of socioeconomic status and maternal psychopathology were intrinsically linked to the factors included in the FAI (either directly or measured very closely in time to those included in the FAI) and were omitted as confounders.

We also included intermediate confounders in our models, that is, factors that may be on the causal pathway from the exposure (FAI) to the mediators (emotional/behaviour problems and constipation) (see figure 2) and thus confound the mediate-outcome relationships. These included child developmental delay, maternal bonding score, parental abuse, stool frequency, hard stools, difficult temperament, picky eating, healthy eating, temper tantrums and body mass index (BMI), all measured between ages 30 and 45 months.[6 9–13] Details on the measurement and coding of the confounders and intermediate confounders are given in online supplemental table 1.

### Statistical analysis

We used univariable and multivariable logistic and linear regression to examine the relationships between (1) FAI and daytime soiling, (2) mediators and daytime soiling and (3) FAI and mediators (emotional/behaviour problems and constipation).

We used the gformula (parametric g-computation formula[18]) command in Stata[19] to examine the mediating effect of emotional/behaviour problems and constipation jointly on the association between FAI score and daytime soiling (see figure 2). G-formula is based on the counterfactual framework and allows the decomposition of effects into the total causal effect (TCE), the natural indirect effect (NIE) (via the mediators of interest), and the natural direct effect (NDE) (not via the mediators of interest). G-formula can also estimate marginal ORs for mediation effects with a binary outcome while incorporating intermediate confounders (see online supplemental text for further information). We used the g-formula to decompose the NIE via emotional/behaviour problems and constipation jointly as we did not consider them to be independent. We used a Monte Carlo sample size of 100 000 and we estimated normal-based 95% CIs using standard errors from 250 bootstrap resamples. The proportion mediated (PM) was calculated as $[OR_{NDE} (OR_{NIE} - 1)] / [OR_{NDE} \times OR_{NIE} - 1] \times 100$ however, this is only interpretable when the TCE, NIE and NDE have the same direction of effect.[20]

Analyses were conducted using Stata V.18[21] and R V.4.3.0.[22]

### Missing data

We used multivariate imputation by chained equations under the missing at random assumption to impute missing data[23] in daytime soiling and baseline and intermediate confounders up to the sample with complete data on the FAI score (N=10 033). We also included auxiliary variables that are considered predictive of the missing values, including socioeconomic status indicators and earlier measures of variables (see online supplemental table 2). Missing data were imputed separately in males and females before appending the two datasets together. For both males and females, 50 imputed datasets with 50 iterations were created using the mice package (V.3.16.0)

in R. Estimates were then combined using Rubin's rules[24] (see online supplemental text).

### RESULTS

Table 1 shows the distribution of study variables in the imputed (N=10 033) and complete case samples (N=3366). Any daytime soiling was reported for 7.3% of children. In total, 71.7% of children experienced one or more family adversities, with the mean number of family adversities experienced being 1.9 (SE 0.02) in all children and being higher in those experiencing soiling (2.3, SE 0.10) compared with those without soiling (1.8, SE 0.02; t-test p<0.001). Further details of the study sample and variables are given in the online supplemental text (Results: description of sample) and online supplemental table 3.

Table 2 presents strong evidence that a one-unit increase in the FAI score was associated with increased odds of 12% for daytime soiling at age 7.5 (adjusted OR 1.12, 95% CI 1.07 to 1.17) after adjustment for confounders. We also found that a one-unit increase in the emotional/behaviour problems score at age 4 was associated with an increase in the odds of daytime soiling at age 7.5 (adjusted OR 1.03, 95% CI 1.002 to 1.05). The presence of constipation at age 6 was strongly associated with increased odds of soiling at age 7.5 (adjusted OR 3.34, 95% CI 2.68 to 4.16).

A one-unit increase in the FAI score was associated with an increase in the emotional/behaviour problems score (adjusted beta 0.52, 95% CI 0.47 to 0.58) and increased odds of constipation (adjusted OR 1.07, 95% CI 1.03 to 1.11; online supplemental table 4). There was weak evidence that a one-unit increase in the emotional/behaviour problems score was associated with constipation (adjusted OR 1.01, 95% CI 0.99 to 1.03; online supplemental table 5).

Table 3 shows the TCE, NIE, NDE and the PM of the one-unit increase in FAI on daytime soiling jointly through the mediators emotional/behaviour problems and constipation. There was weak evidence of a mediating effect ($OR_{NIE}$ 1.03, 95% CI 0.99 to 1.07, PM 27%); however, the effects are imprecise with the 95% CI crossing the null. As a sensitivity analysis, we repeated all analyses with the complete-case sample (n=3366; online supplemental tables 6–8). The direction and magnitude of effects were consistent with the imputed data results.

### DISCUSSION

This is the first study to examine mechanisms that could explain the relationship between early childhood adversity and subsequent soiling at primary school age. Using data from a large community-based birth cohort study with a wide range of confounders, we found that children who were exposed to family adversity in the first 2 years of life were at greater risk of soiling at 7.5 years. There was weak evidence that emotional/behaviour problems

**Table 1** Descriptive statistics of study variables for the imputed sample (N=10 033) and complete case sample (N=3366)

| Variable | Eligible sample (N≤10 033) | | N missing from eligible sample (N=10 033)* | Imputed sample (N=10 033) | | Complete case sample (N=3366) | |
|---|---|---|---|---|---|---|---|
| | Mean | SE | N (%) | Mean | SE | Mean | SE |
| FAI (0–13)† | 1.86 | 0.02 | NA | 1.86 | 0.02 | 1.56 | 0.03 |
| Child development level (Z-score) | 0.01 | 0.01 | 2285 (22.8) | 0.01 | 0.01 | −0.01 | 0.02 |
| Maternal bonding score (10–33) | 27.63 | 0.04 | 1701 (17.0) | 27.61 | 0.04 | 27.72 | 0.06 |
| Child difficult temperament (5–25) | 12.43 | 0.05 | 1307 (13.0) | 12.45 | 0.05 | 12.34 | 0.07 |
| Healthy diet (principal component score) | −0.01 | 0.01 | 1345 (13.4) | −0.01 | 0.01 | −0.04 | 0.02 |
| BMI (kg/m2) | 16.23 | 0.02 | 3131 (31.2) | 16.23 | 0.02 | 16.22 | 0.02 |
| Emotional/behaviour problems score (0–40) | 8.77 | 0.05 | 1791 (17.9) | 8.85 | 0.05 | 8.53 | 0.08 |
| | % | SE | N (%) | % | SE | % | SE |
| Child's sex assigned at birth (female) | 48.73 | 0.5 | NA | 48.73 | 0.5 | 49.67 | 0.86 |
| Maternal smoking during pregnancy | | | | | | | |
| None | 79.06 | 0.41 | 121 (1.2) | 78.95 | 0.41 | 84.46 | 0.62 |
| Yes, quit early | 5.14 | 0.22 | | 5.16 | 0.22 | 3.81 | 0.34 |
| Yes, throughout | 15.81 | 0.37 | | 15.89 | 0.37 | 11.74 | 0.55 |
| Marital status | | | | | | | |
| Married | 81.42 | 0.39 | 155 (1.5) | 81.28 | 0.39 | 86.69 | 0.59 |
| Single | 13.74 | 0.35 | | 13.84 | 0.35 | 9.63 | 0.51 |
| Divorced/separated/widowed | 4.84 | 0.22 | | 4.88 | 0.22 | 3.68 | 0.32 |
| Household social class (manual) | 16.67 | 0.38 | 519 (5.17) | 17.27 | 0.39 | 12.33 | 0.57 |
| Preterm birth | 4.73 | 0.21 | NA | 4.73 | 0.21 | 4.43 | 0.35 |
| Ethnicity (non-white) | 3.82 | 0.19 | 154 (1.5) | 3.88 | 0.2 | 2.47 | 0.27 |
| Stool frequency (low) | 4.97 | 0.23 | 1368 (13.6) | 5.14 | 0.24 | 5.53 | 0.39 |
| Hard stools | 27.93 | 0.48 | 1214 (12.1) | 28.08 | 0.48 | 29.11 | 0.78 |
| Parental abuse | 4.57 | 0.23 | 1589 (15.8) | 5.09 | 0.26 | 4.04 | 0.34 |
| Picky eating | 14.79 | 0.38 | 1298 (12.9) | 14.83 | 0.37 | 15.06 | 0.62 |
| Temper tantrums | 15.55 | 0.39 | 1399 (13.9) | 15.98 | 0.4 | 14.05 | 0.6 |
| Constipation at 6 years | 10.27 | 0.35 | 2488 (24.8) | 10.77 | 0.37 | 10.49 | 0.53 |
| Daytime soiling | 6.72 | 0.29 | 2806 (28.0) | 7.3 | 0.3 | 7 | 0.44 |

50 imputed datasets. The complete case sample comprises individuals with data for all variables described.
*The total number of missing participants for each variable.
†The FAI is not normally distributed, the median (IQR) is 1 (0–3).
BMI, body mass index; FAI, Family Adversity Index; NA, none missing.

and constipation mediated this relationship. Notably, we found that early exposure to family adversity increased the risk of subsequent constipation at age 6. Emotional/behaviour problems at age 4 were weakly associated with constipation at age 6. Consistent with previous studies, we found that the presence of constipation was strongly associated with an increased risk of subsequent soiling.

## Comparison with previous findings and potential mechanisms

We found strong evidence that early exposure to family adversity increased the risk of subsequent soiling and constipation. Family-related stressors have previously been linked to constipation in cross-sectional studies.[5 25] There is evidence of the effect of stress on bowel functioning through the gut-brain axis; a bidirectional communication system including the neural, endocrine, microbial and immune system that connects the two.[26] For example, modulation of bowel motility, secretion and mucosal responses is controlled by the brain.[27] Development of the gut-brain axis during early infancy and childhood may be impacted by psychosocial factors including stressful events and adversity[26] and may be a plausible mechanism through which early family adversity increases risk of later constipation and soiling.

We found only weak evidence for the association between emotional/behavioural problems score at age 4 and constipation at age 6. This is contrary to a previous

**Table 2** Results for the associations between family adversity index and mediators with daytime soiling in the imputed data sample (N=10 033)

| Outcome: daytime soiling | Univariable | | | Multivariable | | |
|---|---|---|---|---|---|---|
| | OR | 95% CI | P value | OR | 95% CI | P value |
| Exposure | | | | | | |
| FAI | 1.11 | 1.06 to 1.16 | <0.001 | 1.12 | 1.07 to 1.17 | <0.001 |
| Emotional/behaviour problems* | 1.07 | 1.05 to 1.09 | <0.001 | 1.03 | 1.00 to 1.05 | 0.04 |
| Constipation*† | 3.32 | 2.69 to 4.10 | <0.001 | 3.34 | 2.68 to 4.16 | <0.001 |

All multivariable models adjusted for baseline confounders (sex, ethnicity, preterm birth, household social class, marital status and smoking during pregnancy). Analysis performed in 50 imputed datasets. Parameter estimates are based on a one-unit increase in FAI, a one-unit increase in emotional/behaviour problems score and the comparison between yes vs no for constipation.
*Models additionally adjusted for intermediate confounders (child developmental delay, maternal bonding score, parental abuse, stool frequency, hard stools, difficult temperament, picky eating, healthy eating, temper tantrums and BMI), and FAI.
†Model additionally adjusted for emotional/behaviour problems.
BMI, body mass index; FAI, Family Adversity Index.

study using ALSPAC data, which reported strong evidence of associations between emotional/behavioural problems at 3.5 years and trajectories of constipation and soiling from age 4 to 10 years.[6] However, the current study uses a measure of constipation at a single time point, compared with the previous study which examined trajectories comprising repeated measures of soiling and constipation across childhood.

We found only weak evidence that emotional/behaviour problems and constipation mediated the relationship between family adversity and children's soiling. It is possible that family adversity is linked to an increased risk of childhood soiling through other causal pathways. For example, inadequate toilet training practices have been linked to an increased risk of bowel disorders.[28] ALSPAC did not collect data on the strategies employed by parents during toilet training but did collect data on age at initiation of toilet training. However, there was little evidence that age at initiation of toilet training was associated with constipation and/or soiling at primary school age.[11] Further research is needed to examine toilet training practices as possible mediators of the link between family adversity and soiling.

Data were not available on important factors at preschool age that are associated with the development of childhood constipation, including stool withholding and toileting refusal. However, our analysis did include hard stools and low stool frequency at age 2½ years as intermediate confounders. These problems can be both causes and consequences of stool withholding and toileting refusal. It has previously been argued that stool withholding is unlikely to be caused primarily by emotional/behaviour problems, since painful defecation usually precedes the onset of stool toileting refusal.[29] However, earlier studies based on small clinic samples have found evidence for more difficult temperament traits in children who are difficult to toilet train, and in children with stool toileting refusal.[30 31] There is also evidence that autistic traits and ADHD are associated with soiling, and it is suggested that this could be due to sensory issues which lead to stool withholding and constipation.[32 33]

At primary school age, the focus of this study, other causes of stool withholding include limited access to toilets, hygiene concerns and perceived lack of privacy or safety of school toilets.[34] Further research is needed to determine the causes of stool toileting refusal in children at different ages and to advance understanding of factors that cause lasting vulnerability to soiling. Identification of biopsychosocial factors on the causal pathway from adversity to soiling could lead to the development of novel therapeutic targets.

**Table 3** Results for the mediation of family adversity index and daytime soiling via emotional/behaviour problems and constipation in the imputed data sample (N=10 033)

| Outcome | Total causal effect | | Natural indirect effect | | Natural direct effect | | |
|---|---|---|---|---|---|---|---|
| | OR | 95% CI | OR | 95% CI | OR | 95% CI | PM% |
| Daytime soiling | 1.10 | 1.04 to 1.17 | 1.03 | 0.99 to 1.07 | 1.07 | 1.01 to 1.14 | 27 |

Mediation models were fitted with emotional/behaviour problems and constipation as the mediators of interest; child developmental delay, maternal bonding score, parental abuse, stool frequency, hard stools, difficult temperament, picky eating, healthy eating, temper tantrums, and BMI were considered intermediate confounders. All paths were adjusted for baseline confounders: sex, ethnicity, preterm birth, household social class, marital status, and smoking during pregnancy. Analysis performed in 50 imputed datasets.
PM, proportion mediated (%).

## Strengths and limitations

Major strengths of this study include the use of data from a large birth cohort, the prospective design, the availability of data on a wide range of relevant confounders and intermediate confounders, and the use of causal mediation methods to examine causal pathways between family adversity in early childhood and subsequent risk of soiling at school age.

We did not restrict our analysis to children who met clinical diagnostic criteria for faecal incontinence; therefore, the study findings apply to children with soiling in the community and not just those with the most severe soiling.[11] This is a community-based sample, and therefore, the proportion of participants who experienced soiling at high frequencies is small compared with clinical samples (0.83%). We therefore examined the presence vs absence of any soiling and did not further categorise by frequency because this would have resulted in very small group sizes and lack of precision in our estimates. It is important to note that we found robust associations between family adversity and soiling, even when examining soiling that did not meet the criteria for clinical diagnosis.[35] We also acknowledge that any level of soiling, even if infrequent, is distressing for children at primary school age and can lead to peer rejection and social isolation.[36] Parents were not asked about the duration of constipation and soiling at each time point, but the repeated measures of constipation and soiling in the ALSPAC cohort indicate that constipation and soiling are chronic problems.[11]

The complete-case sample had a lower proportion of participants with socioeconomic disadvantage (eg, manual social class and divorced/separated/widowed marital status) and FAI than the wider ALSPAC cohort, which may limit the internal validity of our findings. We therefore used multiple imputation to address possible bias due to missing data and report all results as recommended by Sterne *et al*.[37]

Reliance on parental reports of constipation is a possible limitation. Parents were asked to report if their child had suffered from constipation in the past 12 months and if they had seen a doctor but were not asked about the frequency or severity of constipation. Seeing a doctor for constipation is not necessarily an indicator of severity but might be determined by factors such as parental health literacy. Underreporting of constipation is possible since some parents may have been unaware that their child is constipated. However, the fact that the prevalence of constipation in ALSPAC is similar to other population-based studies[38] provides reassurance concerning the validity of the parental reports. The use of parental reports of soiling and the specific phrasing of the question ('how often does your child dirty their pants during the day') could have led to the possibility of parents responding 'yes' to the soiling question due to their child's inadequate wiping following a bowel movement (rather than an episode of faecal incontinence). It is notable, however, that some parents stated in the questionnaire that their child had soiled his/her pants only because of poor wiping or due to suffering an episode of diarrhoea and in these cases they did not place their child in the soiling category.[6]

We did not differentiate between retentive and non-retentive subtypes of soiling. Clinical studies, however, estimate that over 80% of children who experience functional faecal incontinence have underlying constipation.[35] We also did not examine nocturnal soiling as an outcome at age 7.5, but the prevalence was very low at this age in the ALSPAC cohort (0.8%).

## Implications

Clinicians should be aware that exposure to family adversity in the early years places children at greater risk of subsequent constipation and soiling. Screening for adversities could be conducted at routine contacts with health visitors, general practitioners and in paediatric bladder and bowel clinics. Increased resources need to be directed at providing support and interventions for families who are experiencing adversities to reduce the impact on children's health and development.

Constipation in preschool children is still not universally understood and managed by primary care practitioners in much of the UK, despite National Institute for Health and Care Excellence (NICE) guidance and a wide evidence base for proactive treatment.[39] There are still some areas in the UK without bladder and bowel services for children and other areas where services do exist but are restricted to seeing children aged over 5 years or are inadequately resourced, resulting in long wait times for children to receive specialist assessment and support. This is a serious concern because chronic constipation and soiling in children have adverse impacts on their quality of life and psychosocial functioning.[39] Our findings confirm the importance of training for all healthcare professionals, early years practitioners and parents/carers on signs and symptoms of constipation in children from birth onwards to enable prompt intervention before it becomes chronic.

**Author affiliations**
[1]Centre for Academic Child Health, Population Health Sciences, Bristol Medical School, University of Bristol, Bristol, UK
[2]Centre for Academic Mental Health, Population Health Sciences, Bristol Medical School, University of Bristol, Bristol, UK
[3]Medical Research Council Integrative Epidemiology Unit, Population Health Sciences, Bristol Medical School, University of Bristol, Bristol, UK
[4]Children's Nurse Specialist, Bladder and Bowel UK, The Toilet Training Team, UK
[5]ERIC, The Children's Bowel and Bladder Charity, 36 Old School House, Kingswood Foundation, Brittania Rd, Bristol, UK
[6]Paediatric Gastroenterology Department, Bristol Royal Hospital for Children, Bristol, UK

**Acknowledgements** We are extremely grateful to all the families who took part in this study, the midwives for their help in recruiting them and the whole ALSPAC team, which includes interviewers, computer and laboratory technicians, clerical workers, research scientists, volunteers, managers, receptionists and nurses.

**Contributors** CJ, JH and KB: conceptualisation; CJ and JH: funding acquisition; KB, CJ and JH: analytical plan; KB, JH and CJ: methodology; KB: formal analysis; KB, CJ, JH, DR, JR and CS: interpretation of findings; KB and CJ: writing original

draft; KB, CJ, JH, DR, JR and CS: writing review and editing. This publication is the work of the authors and KB will serve as guarantors for the contents of this paper.

**Funding** This work is supported by funding from the Medical Research Council (grant ref: MR/V033581/1: Mental Health and Incontinence). JH is a member of the MRC Integrative Epidemiology Unit at the University of Bristol (MC_UU_00011/7). The UK Medical Research Council and Wellcome (grant ref: 217065/Z/19/Z) and the University of Bristol provide core support for ALSPAC. A comprehensive list of grants funding is available on the ALSPAC website: http://www.bristol.ac.uk/alspac/external/documents/grant-acknowledgements.pdf.

**Competing interests** None declared.

**Patient and public involvement** Patients and/or the public were not involved in the design, or conduct, or reporting, or dissemination plans of this research.

**Patient consent for publication** Not applicable.

**Ethics approval** Ethical approval for the study was obtained from the ALSPAC Ethics and Law Committee and the Local Research Ethics Committees. Research ethics committee approval references can be found at http://www.bristol.ac.uk/alspac/researchers/research-ethics/. Initial approval references were granted by Bristol and Weston Health Authority (E1808), Southmead Health Authority (49/89), and Frenchay Health Authority (90/8). Informed consent for the use of all data collected was obtained from participants following the recommendations of the ALSPAC Ethics and Law Committee at the time. Participants can contact the study team at any time to retrospectively withdraw consent for their data to be used. Study participation is voluntary and during all data collection sweeps, information was provided on the intended use of data.

**Provenance and peer review** Not commissioned; externally peer reviewed.

**Data availability statement** Data are available on reasonable request. The ALSPAC data underlying this article are available on request from the ALSPAC Executive Committee for researchers who meet the criteria for access to confidential data (https://bristol.ac.uk/alspac/researchers/access/). Code to conduct these analyses can be found on https://github.com/burrowsk/Family-adversity-constipation-soiling.

**ORCID iDs**
Kimberley Burrows https://orcid.org/0000-0002-3208-0389
Jon Heron https://orcid.org/0000-0001-6199-5644
Christine Spray https://orcid.org/0000-0002-2885-9146

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
