## [Reviewer comments · BMJ Paediatrics Open]

ARTICLE DETAILS

TITLE (PROVISIONAL)	Examining causal pathways between family adversity and soiling: a prospective cohort study
AUTHORS	Burrows, Kimberley; Heron, Jon; Richardson, Davina; Rayner, Juliette; Spray, Chris; Joinson, Carol

VERSION 1 - REVIEW

REVIEWER NAME	Rebecca Pollard
REVIEWER AFFILIATION	King's IoPPN
REVIEWER CONFLICT OF INTEREST	N/A
AI DISCLOSURE	No
DATE REVIEW RETURNED	21-Sep-2025

GENERAL COMMENTS	The manuscript presents interesting and potentially novel findings regarding the association between cumulative childhood adversity burden and soiling in infancy, and particularly the role of emotional problems and constipation as mediators in this relationship. This research is important, as it may help clarify underlying mechanisms in the relationship between childhood adversity and soiling, and guide future intervention approaches. I have several points for the authors to consider: - Novelty: From the introduction, the authors have described that this study provides new insights into the association between cumulative adversity and soiling, with mediation analyses addressing potential mechanisms. It would strengthen the paper if the authors could more clearly highlight the novelty of these contributions and how they add to existing literature.- Operationalization of soiling: The analysis considered soiling in a binary fashion. While I understand from the discussion that this decision was partly due to the low percentage of participants reporting soiling (~7%), it would be helpful to clarify this more explicitly in the methods. In addition, could the authors reflect on how the results might differ if frequency of soiling were analysed (e.g., more vs. less frequent cases)?- Mediation model: Emotional problems and constipation were entered jointly as mediators. Further justification for this modelling decision would strengthen the paper, especially as the data from this sample indicate that these two variables were not highly correlated. It would be useful to explain why both were modelled together, and whether separate models were also considered.- Sex differences: The analyses controlled for sex, but there is emerging literature suggesting that males and females may respond differently to childhood adversity, both in terms of emotional and somatic outcomes. It would add value if the authors could explore whether stratified analyses by sex might yield similar or divergent patterns.
--

	- Ethics statement: The ethics section should include the name of the approving body and the corresponding approval/reference number. If no number was issued, this should be clarified.
--	--

VERSION 1 – AUTHOR RESPONSE

Reviewer: 1

Dr. Rebecca Pollard, King's IoPPN

Comments to the Author

The manuscript presents interesting and potentially novel findings regarding the association between cumulative childhood adversity burden and soiling in infancy, and particularly the role of emotional problems and constipation as mediators in this relationship. This research is important, as it may help clarify underlying mechanisms in the relationship between childhood adversity and soiling, and guide future intervention approaches.

We thank the reviewer for taking the time to review our paper and for the suggestions to improve our manuscript.

I have several points for the authors to consider:

- Novelty: From the introduction, the authors have described that this study provides new insights into the association between cumulative adversity and soiling, with mediation analyses addressing potential mechanisms. It would strengthen the paper if the authors could more clearly highlight the novelty of these contributions and how they add to existing literature.

Thank you for highlighting this. We have now added further information to more strongly highlight the novelty of our study. Specifically, we have included:

In the Key Messages:

“What this study adds?”

- This is the first study to examine mechanisms explaining the relationship between adversity and soiling using causal mediation methods in a large community-based birth cohort study.
- The total burden of family adversity (encompassing facets of psychopathology, discord and events) during infancy increases the odds of soiling at primary school age.
- There is weak evidence that emotional/behaviour problems and constipation mediate the relationship between family adversity and soiling.”

In the Discussion

“This is the first study to examine mechanisms that could explain the relationship between early childhood adversity and subsequent soiling at primary school age. Using data from a large community-based birth cohort study with a wide range of confounders, we found that children who were exposed to family adversity in the first two years of life were at greater risk of soiling at 7.5 years. There was weak evidence that emotional/behaviour problems and constipation mediated this relationship. Notably, we found that early exposure to family adversity increased the risk of subsequent constipation at age 6. Emotional/behaviour problems at age 4 were weakly associated

with constipation at age 6. Consistent with previous studies, we found that the presence of constipation was strongly associated with an increased risk of subsequent soiling.”

- Operationalization of soiling: The analysis considered soiling in a binary fashion. While I understand from the discussion that this decision was partly due to the low percentage of participants reporting soiling (~7%), it would be helpful to clarify this more explicitly in the methods. In addition, could the authors reflect on how the results might differ if frequency of soiling were analysed (e.g., more vs. less frequent cases)?

This is a community-based sample, and therefore, the proportion of participants who experienced soiling at high frequencies (e.g, soiling every day) is small (only 0.1%) compared with clinical samples. We therefore examined the presence versus absence of any soiling and did not further categorise by frequency because this would have resulted in very small group sizes (only ~ 0.8% reported soiling multiple times per week) and lack of precision in our estimates. It is important to note that we found robust associations between family adversity and soiling, even when examining soiling that did not meet the criteria for clinical diagnosis. We also acknowledge that any level of soiling, even if infrequent is distressing for children at primary school age and can lead to peer rejection and social isolation [<https://pubmed.ncbi.nlm.nih.gov/29228510/>].

We have added the following to the methods to help clarify the decision to derive the outcome as we did:

“When the study children were 7 years 8 months (SD = 0.14) (hereafter referred to as 7.5 years) mothers were asked “how often does your child dirty their pants during the day”. Options included “Never”; “Occasional accidents but less than once a week”; “About once a week”; “2-5 times a week”; “Nearly every day”; and “More than once a day”. After exploring the proportions of respondents for the more frequent soiling categories (only 0.83% responded with 2-5 times a week or more), we chose to derive a binary variable to indicate any daytime soiling by assigning a value of 0 to “Never” and 1 to all other categories. Further information for coding daytime soiling is given in Supplementary Table 1.2”

- Mediation model: Emotional problems and constipation were entered jointly as mediators. Further justification for this modelling decision would strengthen the paper, especially as the data from this sample indicate that these two variables were not highly correlated. It would be useful to explain why both were modelled together, and whether separate models were also considered.

We planned a priori to explore the two mediators jointly as we hypothesised that emotional and behaviour problems are related to constipation, with both being on the causal pathway from early childhood adversity to subsequent soiling at primary school age. In our previous studies we found evidence that behaviour/emotional problems were associated with increased odds of constipation and soiling via a latent class analysis of trajectories of constipation and soiling. Specifically, behavioural and emotional problems were most strongly associated with the “constipation with soiling” latent class [<https://pubmed.ncbi.nlm.nih.gov/29748737/>]. However, we have explored both constipation and soiling as single observations at specific timepoints in order to establish a temporal prospective ordering in a mediation analysis. As you point out, after adjustment for baseline confounders we found that the two variables were not highly correlated. However, we implemented our analysis plan based on previous evidence and our hypothesis and chose not to implement post hoc analysis to explore the factors independently.

We have added the following to the methods to strengthen the justification of jointly modelling the two mediators:

“Mediators: emotional/behaviour problems and constipation

We chose emotional/behavioural problems and constipation as mediators as these are potentially modifiable and could be targeted in interventions aimed at reducing the risk of soiling [7,15–19]. Furthermore, there is evidence that emotional/behavioural problems are associated with constipation and soiling [22,23] and so we considered the two mediators jointly within our causal mediation models (see Figure 2).

When the study child was 4 years (SD = 0.12), mothers completed...”

- Sex differences: The analyses controlled for sex, but there is emerging literature suggesting that males and females may respond differently to childhood adversity, both in terms of emotional and somatic outcomes. It would add value if the authors could explore whether stratified analyses by sex might yield similar or divergent patterns.

Thank you for your comment and we agree that there is emerging literature suggesting differential response to adversity by sex. We find little evidence for an interaction by sex for the adversity exposure and each of the mediator pathways (likelihood ratio test for interaction by sex P values 0.08 (emotional and behaviour problems as the outcome) and 0.36 (constipation as the outcome)), nor for the adversity and soiling association (LRR P value for interaction by sex 0.70), where all analyses were adjusted for relevant confounders and conducted in the complete case sample. We therefore do not anticipate divergent patterns of associations when stratifying by sex.

- Ethics statement: The ethics section should include the name of the approving body and the corresponding approval/reference number. If no number was issued, this should be clarified.

We have further elaborated on our ethics statement which now includes:

“Ethical approval for the study was obtained from the ALSPAC Ethics and Law Committee and the Local Research Ethics Committees. Research ethics committee approval references can be found at <http://www.bristol.ac.uk/alspac/researchers/research-ethics/>. Initial approval references were granted by Bristol and Weston Health Authority (E1808), Southmead Health Authority (49/89), and Frenchay Health Authority (90/8). Informed consent for the use of all data collected was obtained from participants following the recommendations of the ALSPAC Ethics and Law Committee at the time. Participants can contact the study team at any time to retrospectively withdraw consent for their data to be used. Study participation is voluntary and during all data collection sweeps, information was provided on the intended use of data.”